

# Population structure of *Bathymodiolus manusensis*, a deep-sea hydrothermal vent-dependent mussel from Manus Basin, Papua New Guinea

Andrew D. Thaler[1,2], William Saleu[1,3], Jens Carlsson[4], Thomas F. Schultz[1] and Cindy L. Van Dover[1]

[1] Division of Marine Science and Conservation, Nicholas School of the Environment, Duke University, Beaufort, NC, USA
[2] Blackbeard Biologic: Science and Environmental Advisors, St. Michaels, MD, USA
[3] BETA Scientific, Port Moresby, Papua New Guinea
[4] Area52 Research Group, School of Biology and Environmental Science, Earth Institute, University College Dublin, Dublin, Ireland

Corresponding author
Andrew D. Thaler,
andrew@blackbeardbiologic.com,
andrew.david.thaler@gmail.com

## ABSTRACT

Deep-sea hydrothermal vents in the western Pacific are increasingly being assessed for their potential mineral wealth. To anticipate the potential impacts on biodiversity and connectivity among populations at these vents, environmental baselines need to be established. *Bathymodiolus manusensis* is a deep-sea mussel found in close association with hydrothermal vents in Manus Basin, Papua New Guinea. Using multiple genetic markers (*cytochrome C-oxidase subunit-1* sequencing and eight microsatellite markers), we examined population structure at two sites in Manus Basin separated by 40 km and near a potential mining prospect, where the species has not been observed. No population structure was detected in mussels sampled from these two sites. We also compared a subset of samples with *B. manusensis* from previous studies to infer broader population trends. The genetic diversity observed can be used as a baseline against which changes in genetic diversity within the population may be assessed following the proposed mining event.

## INTRODUCTION

Hydrothermal vents support large, endemic communities fueled by chemoautotrophic primary production (*Gage & Tyler, 1991*; *Van Dover, 2000*), in contrast to the relatively low-biomass found on the deep seafloor. In Southwest Pacific back-arc basins, active vents are patchily distributed and subject to local disturbances, including the waxing and waning of hydrothermal flow on short time scales and cessation of flow on millennial timescales (*Van Dover, 2000*; *Vrijenhoek, 2010*). Species demographics may be driven as much by stochastic processes related to disturbance as by response to changing environmental conditions or other ecological phenomena (*Vrijenhoek, 2010*; *Thaler et al., 2014*). In general,

hydrothermal vent communities are thought to be more resilient to disturbance compared to other deep-sea ecosystems (*Van Dover, 2014*).

Deep-sea hydrothermal vents are increasingly being explored for potential mineral extraction (*Van Dover, 2010*). Almost 20% of all known global vent fields currently fall within mining exploration leases (*Beaulieu et al., 2013*). Establishing baselines for the diversity and connectivity of vent systems is a necessary first step in effective environmental management regimes (*Collins et al., 2013*). As vents become targets for mineral extraction, managers will need to assess regional biodiversity and connectivity and potential cumulative impacts of multiple mining events in a region (*Boschen et al., 2016*; *Van Dover, 2014*) and design refugia to mitigate the impacts of mining on the vent ecosystem (*Collins, Kennedy & Van Dover, 2012*).

*Bathymodiolus manusensis* is a deep-sea mussel found at hydrothermal vents in the Manus Basin, Papua New Guinea. It commonly occurs around low-temperature diffuse-flow vent sites on the periphery of active hydrothermal chimneys (*Hashimoto & Furuta, 2007*). Though *B. manusensis* shares close affinity with other bathymodiolin mussels in Lau and North Fiji Basins (*B. brevior*), *B. manusensis* is primarily found within Manus Basin (*Hashimoto & Furuta, 2007*; *Kyuno & Shintaku, 2009*) and has been reported from a few sites in Lau Basin (*Miyazaki et al., 2004*), as well as off the coast of New Zealand (*Miyazaki et al., 2010*). *B. manusensis* is one of several habitat-forming mollusks that host chemoautotrophic endosymbionts and derive chemical energy from hydrothermal vent effluent in Manus Basin (*Galkin, 1997*). While other endosymbiont-hosting species at Manus Basin vents, such as *Ifremeria nautilei* and *Alviniconcha* spp. (*Bouchet & Waren, 1991*; *Kojima et al., 2001*; *Urakawa et al., 2005*), tend to cluster around orifices where vent effluent is most concentrated, *B. manusensis* occupies the periphery of high temperature hydrothermal ecosystems, taking advantage of the less space-restrictive regions around diffuse flow sites (*Kyuno & Shintaku, 2009*).

The Solwara 1 vent site in Manus Basin is licensed for extraction of metals associated with seafloor massive sulfides (*Coffey Natural Systems, 2008*). While *B. manusensis* does not occur at Solwara 1, it is abundant at the neighboring Solwara 8 site (40 km distant) and at the proposed set-aside, South Su (2.5 km distant; *Coffey Natural Systems, 2008*). Previous studies of connectivity in invertebrate taxa at these sites reveal species-specific patterns of connectivity among sites. *Ifremeria nautilei* and *Chorocaris* sp. 2, two endosymbiont-hosting vent species show no signs of genetic differentiation among Solwara 1, Solwara 8, and South Su (*Thaler et al., 2011*; *Thaler et al., 2014*) while significant local differentiation was detected in the vent-associated *Munidopsis lauensis* (*Thaler et al., 2014*). *B. manusensis* has a limited geographic range and is not ubiquitous at active vents in Manus Basin, leading us to anticipate that it might exhibit local-scale genetic differentiation. Because *B. manusensis* is absent from Solwara 1 and because Solwara 1 is situated between Solwara 8 and South Su, we tested the hypothesis that populations from South Su and Solwara 8 are isolated from each other, forming two genetically distinct populations.
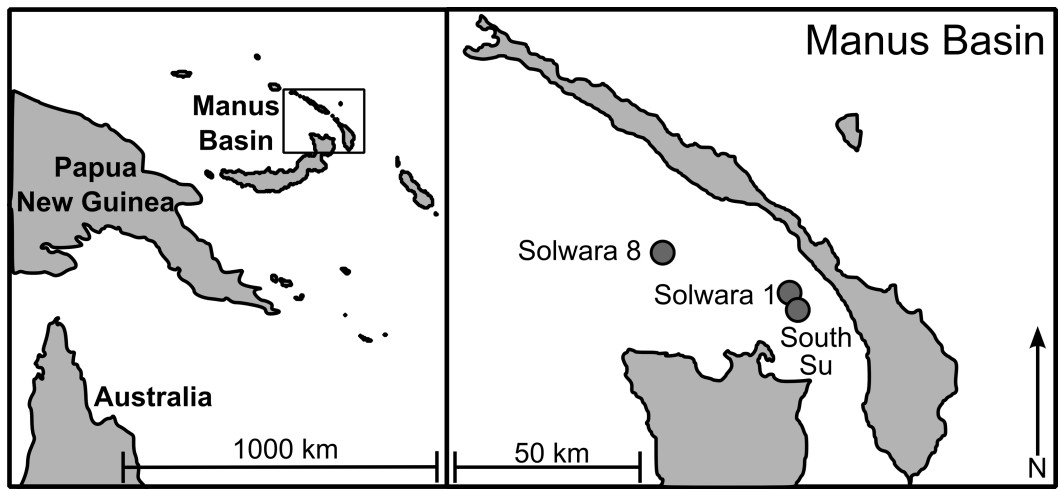

**Figure 1** Sampling locations in Manus Lau Basin. Figure adapted from one originally published in *Thaler et al. (2011)*.

**Table 1** Bathymodiolus manusensis sampling locations in Manus Basin.

| Site | Mound | Latitude | Longitude | Depth (m) |
|------|-------|----------|-----------|-----------|
| Solwara 8 | Mound 1 | 3°43.740′S | 151°40.404′E | 1,720 |
| | Mound 2 | 3°43.824′S | 151°40.458′E | 1,710 |
| South Su | Mound 3 | 3°48.564′S | 152°6.144′E | 1,300 |
| | Mound 4 | 3°48.492′S | 152°6.186′E | 1,350 |

## MATERIALS AND METHODS

### Sample collection and DNA extraction

As part of a larger study looking at multi-species biodiversity and population structure within Manus Basin, *Bathymodiolus manusensis* were collected from two hydrothermal vent sites (Solwara 8 and South Su; Fig. 1) during the *M/V Nor Sky* research campaign (June–July 2008; Chief Scientist: S Smith) using an ST200 ROV modified for biological sampling. For *COI* analyses, *B. manusensis* were analyzed from two discrete sulfide mounds at each site (Table 1), with 10–43 individuals per mound (Table 2). For microsatellite analyses, up to 142 individuals per locus were analyzed. All sampling was undertaken with the permission of the government of Papua New Guinea and did not involve endangered or protected species.

Mantle tissue was dissected from each individual and preserved in 95% ethanol prior to DNA extraction. Genomic DNA was isolated using a standard Chelex-Proteinase-K extraction (10–30 mg tissue digested with 120 μg Proteinase K (Bioline, Taunton, MA, USA) in 600 μl 10% Chelex-100 resin (Bio-Rad, Hercules, CA, USA)) overnight at 60 °C, heated to 100 °C for 15 min, and centrifuged at 10,000 rpm for 5 min; (*Walsh, Metzger & Higuchi, 1991*). Extracted DNA was stored at 4 °C until amplification and archived at −20 °C.

**Table 2** *Bathymodiolus manusensis.* **Summary statistics for *COI* sequences (409 bp) from Manus Basin.**

| Location | $N$ | $H$ | $Hd$ | $F_S$ |
|---|---|---|---|---|
| Manus Basin (total) | 100 | 20 | 0.52 | **−25.60** |
| Solwara 8 | 47 | 9 | 0.45 | **−7.29** |
| Mound 1 | 34 | 9 | 0.46 | **−8.12** |
| Mound 2 | 13 | 4 | 0.42 | −1.66 |
| South Su | 53 | 16 | 0.59 | **−18.32** |
| Mound 3 | 10 | 3 | 0.38 | −1.16 |
| Mound 4 | 43 | 14 | 0.63 | **−13.97** |

**Notes.**

$N$, number of individuals; $H$, number of haplotypes; $Hd$, haplotype diversity; $F_S$, Fu's $F_S$.
Significant Fu's $F_S$ indicated in bold.

## *COI* sequencing and analysis

*Bathymodiolus manusensis* mitochondrial *COI* fragments were amplified using the following reaction conditions: 10–100 ng of DNA template was combined with 2 μL 10× PCR buffer (200 mM Tris, pH 8.8; 500 mM KCl; 0.1% Triton X-100; 0.2 mg/ml BSA), 2 mM MgCl$_2$, 0.2 mM dNTPs, 0.5 μM LCOI1490 and 0.5 μM HCOI2198 primers (*Folmer et al., 1994*), and 1 unit of Taq polymerase in a 20 μL reaction with the following PCR protocol: initial melting temperature of 94 °C for 240 s; 35 cycles of 94 °C for 15 s, 48 °C for 15 s, 72 °C for 30 s; and a final extension of 72 °C for 300 s. Products were stored at 4 °C until purification.

Fourteen μl of PCR product was incubated with 0.2 μl 10× ExoAP buffer (500 mM Bis-Tris, 10 mM MgCl2, 1 mM ZnSO4), 0.05 μl Antarctic Phosphatase (New England Biolabs, Ipswich, MA, USA), 0.05 μl Exonuclease I (New England Biolabs, Ipswich, MA, USA) at 37 °C for 60 min followed by 85 °C for 15 min to remove unincorporated nucleotides. Sequencing reactions were executed with BigDye Terminator v3 reactions (Applied Biosystems, Foster City, CA, USA). AMPure magnetic beads (Agencourt; Morrisville, NC, USA) were used to remove excess dye, products were analyzed on an ABI 3730xl DNA Analyzer (Applied Biosystems International, Rotkreuz, Switzerland), and chromatograms were edited using CodonCode Aligner (version 3.7.1; CodonCode Corporation, Dedham, MA, USA). Consensus sequences were compared against the NCBI GenBank database to confirm identity when available (*Benson, 1997*) and sequence alignments were constructed using the MUSCLE alignment algorithm (*Edgar, 2004*) implemented in CodonCode Aligner. Representative sequences of dominant haplotypes were deposited in GenBank (accession numbers KF498731–KF498847). Full *COI* sequences for each individual are provided as FASTA files (Data S1).

Standard summary statistics, including number of haplotypes ($H$), haplotype diversity ($Hd$), nucleotide diversity ($\pi$), and Fu's $F_S$ were calculated using DnaSP version 5.10.01 (*Librado & Rozas, 2009*). To detect potential cryptic species, maximum-parsimony phylograms of aligned mitochondrial sequences were assembled in MEGA version 5 (10,000 replicates; Tamura 3-parameter substitution model determined by Mega 5: Find Best-Fit Substitution Model; *Tamura et al., 2011*). To visualize potential population structure,

statistical-parsimony networks were assembled in TCS version 1.21 (default settings; *Clement, Posada & Crandall, 2000*). To detect population structure, Arlequin version 3.5.1.2 (*Excoffier, Laval & Schneider, 2005*) was used to estimate pairwise $\varphi_{ST}$. Sequential Bonferroni was used in all appropriate comparisons to correct for multiple tests (*Rice, 1989*).

Additional *Bathymodiolus manusensis* samples were identified in NCBI GenBank (accession numbers KU597590.1 through KU597592.1 from PACMANUS in Manus Basin (*Assié et al., 2016*); AB101431.1 through AB101434.1 from PACMANUS in Manus Basin (*Miyazaki et al., 2004*); AB257539.1, AB257541.1, and AB257543.1 from Lau Basin (*Miyazaki et al., 2010*); and AB255739.1, AB255740.1, AB255741.1, and AB255742.1 from offshore New Zealand (*Miyazaki et al., 2010*)). A Neighbor-Joining tree (10,000 bootstrap replicates) was assembled from three sequenced individuals from Solwara 8, three individuals from South Su, as well as seven individuals from PACMANUS, three individuals from Lau Basin, and four individuals from offshore New Zealand GenBank sequences using MEGA 7 (*Kumar, Stecher & Tamura, 2016*).

## Microsatellite genotyping and statistical analyses

Eight microsatellite markers (*Bm17, Bm22, Bm23, Bm53, Bm63, Bm76, Bm81, Bm83*) were amplified from *Bathymodiolus manusensis* in Manus Basin following methods reported in (*Schultz et al., 2010*). To test whether these markers provided sufficient power to evaluate the null hypothesis of genetic homogeneity, models of the dataset were implemented in POWSIM (Settings based on observed allele distributions, Supplement 1; *Ryman & Plam, 2006*). Full microsatellite genotypes for each individual are provided as GENPOP files (Data S2).

Standard summary statistics, including divergence from expected Hardy-Weinberg Equilibrium (HWE) and allelic richness were assessed using GENEPOP (default settings; version 4.0; *Rousset, 2008*) and Microsatellite Analyzer (version 4.05; *Dieringer & Schlötterer, 2003*), respectively. Permutation tests were used to determine significant variation in allelic richness (*F*-stat; default settings; version 2.9.3.2; *Goudet, 1995*). MicroChecker (version 2.2.3; 1,000 randomizations; *Van Oosterhout et al., 2004*) was used to detect the potential presence of null alleles, stutter, and large allele dropout. To test for the potential influence of selection, loci were screened using LOSITAN (25,000 simulations; IA and SMM; *Antao et al., 2008*; *Beaumont & Nichols, 1996*).

Pairwise genetic differentiation ($F_{ST}$) between aggregations, sites, and basins was analyzed using Microsatellite Analyzer. Alpha levels were adjusted via Sequential Bonferroni to correct for multiple tests (*Rice, 1989*). Structure version 2.3.3 (admixture model, sampling locations as prior distributions; *Pritchard, Stephens & Donnelly, 2000*) was used to visualize potential population structure. Analyses were conducted with a 1,000,000 step burn-in, 10,000,000 repetitions, and three replicates per level from $K = 1$–7. Effective population size was estimated based on microsatellite linkage-disequilibrium using LDNe (default parameters; *Waples & Do, 2008*).
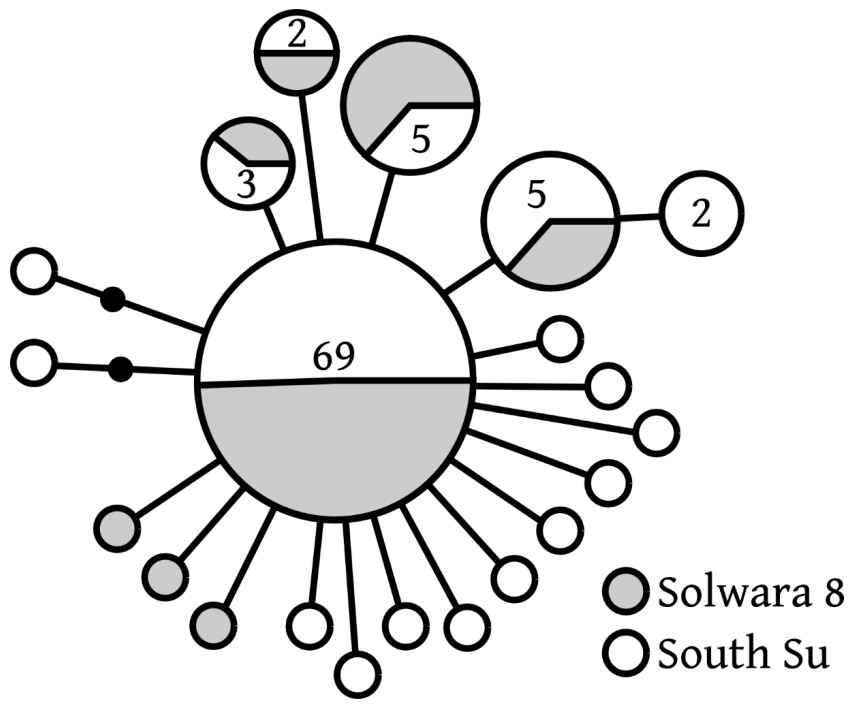

**Figure 2** *Bathymodiolus manusensis*. **Statistical parsimony network for *COI* haplotypes from samples collected at Solwara 8 and South Su, Manus Basin.** Large circles represent a single individual unless noted on the figure. Small black circles represent inferred haplotypes not observed in this data set. Solwara 8 represented by gray circles. South Su represented by white circles. Each node represents one base pair difference.

## RESULTS

Of 20 *COI* haplotypes (409 bp) identified, five were shared at both sites, three were only found in Solwara 8 samples, and 13 were only found at South Su. *Bathymodiolus manusensis* from Solwara 8 (47 individuals) and South Su (53 individuals) in Manus Basin (Table 2) comprised a single haplogroup, based on *COI* analysis. A maximum of 5 base-pair mutations separated the most divergent haplotypes (Table 2, Fig. 2). The statistical parsimony network for *B. manusensis* has a wheel-and-spoke topology, with a single central dominant haplotype and numerous low-abundance secondary haplotypes (Fig. 2). The dominant haplotype is roughly evenly distributed among both sites and all relatively abundant haplotypes ($n \geq 3$) occur at both Solwara 8 and South Su (Fig. 2). Fu's $F_S$ values for *COI* sequence data were significantly negative for samples pooled from both sites, as well as within sites and at Mound 1 (Solwara 8) and Mound 4 (South Su; Table 2).

Neighbor-joining phylogenetic analysis of a 370-base pair *COI* segment shared among *Bathymodiolus manusensis* from neighboring sites and basins indicated low variability across sites, with samples from within Manus Basin being closely related to each other, while samples from Lau Basin and offshore New Zealand were basal to all Manus Basin samples (Fig. S1).

Eight microsatellite loci were amplified from *Bathymodiolus manusensis* (35–142 individuals per site; Table 3). Alleles per locus ranged from 3 to 20 (mean = 10). Allelic

**Table 3  Summary statistics for eight microsatellite loci amplified from *Bathymodiolus manusensis* from Manus Basin.**

|  |  | Bm17 | Bm22 | Bm23 | Bm53 | Bm63 | Bm76 | Bm81 | Bm83 |
|---|---|---|---|---|---|---|---|---|---|
| Solwara 8 | $n$ | 140 | 137 | 126 | 133 | 91 | 136 | 129 | 142 |
|  | a | 3 | 17 | 8 | 5 | 20 | 6 | 16 | 12 |
|  | $Rs$ | 3.00 | 15.74 | 6.04 | 4.66 | 13.85 | 5.70 | 14.14 | 8.75 |
|  | as | 264–368 | 236–284 | 238–270 | 243–264 | 198–262 | 189–206 | 197–245 | 206–224 |
|  | $H_O$ | 0.24 | 0.91 | 0.55 | **0.44** | 0.78 | 0.64 | 0.86 | 0.58 |
|  | $H_E$ | 0.29 | 0.89 | 0.61 | **0.57** | 0.85 | 0.65 | 0.86 | 0.55 |
| South Su | $n$ | 65 | 63 | 53 | 61 | 35 | 58 | 60 | 61 |
|  | a | 3 | 18 | 5 | 4 | 13 | 7 | 13 | 8 |
|  | $Rs$ | 3.00 | 18.00 | 5.00 | 4.00 | 13.00 | 7.00 | 13.00 | 8.00 |
|  | as | 264–268 | 228–287 | 238–265 | 251–264 | 198–263 | 189–210 | 188–258 | 210–223 |
|  | $H_O$ | 0.29 | 0.89 | 0.62 | 0.43 | 0.94 | 0.71 | 0.73 | 0.54 |
|  | $H_E$ | 0.28 | 0.86 | 0.62 | 0.47 | 0.88 | 0.69 | 0.85 | 0.56 |

Notes.

$n$, number of individuals; a, number of alleles; $Rs$, allelic richness; $H_E$, expected heterozygosity; $H_O$, observed heterozygosity; bold, significant heterozygote deficiency.

**Table 4  Pairwise comparisons of *Bathymodiolus manusensis* from two mound each of two sites in Manus Basin.** $F_{ST}$ from microsatellites above the diagonal, $\varphi_{ST}$ from partial *COI* below the diagonal. No pairwise estimates of population differentiation were significant ($P < 0.05$).

|  | Solwara 8 Mound 1 | Solwara 8 Mound 2 | South Su Mound 3 | South Su Mound 4 |
|---|---|---|---|---|
| SW8 Mound 1 | – | 0.00 | 0.00 | 0.00 |
| SW8 Mound 2 | 0.00 | – | 0.00 | 0.00 |
| SSU Mound 3 | 0.00 | 0.02 | – | 0.00 |
| SSU Mound 4 | 0.00 | 0.00 | 0.00 | – |

richness ($Rs$) did not vary significantly among mounds or sites (10,000 permutations, $P > 0.05$; Table 3) and neither balancing nor directional selection was detected at any spatial scale (LOSITAN, $P > 0.05$). Only one marker deviated from Hardy-Weinberg expectations and showed evidence for heterozygote deficiency at Solwara (*Bm53*; Table 3). POWSIM indicated that the sample set has sufficient statistical power to accept or reject the null hypothesis of genetic homogeneity. MicroChecker indicated that null alleles were present at that loci and were responsible for heterozygote deficiencies. As the presence of null alleles has been shown not to severely bias assignment tests (*Carlsson, 2008*), this marker was included in subsequent analyses.

Analysis of Molecular Variance (AMOVA) and pairwise tests for population differentiation ($F_{ST}$ and $\varphi_{ST}$) based on *COI* sequences and microsatellite markers indicated no significant genetic differentiation among *Bathymodiolus manusensis* from Solwara 8 and South Su (Table 4). Assignment tests for combined *COI* and microsatellite data placed all *B. manusensis* into a single population (Structure, $K = 1$, data not shown). Effective population size estimated from microsatellite linkage disequilibrium (LDNe) was functionally infinite.

## DISCUSSION

### Population structure of *Bathymodiolus manusensis* in Manus Basin

*Bathymodiolus manusensis* form a single, coherent population between Solwara 8 and South Su in Manus Basin, Papua New Guinea. No genetic differentiation was detected at any spatial scale using either mitochondrial *COI* or nuclear microsatellite markers. Despite this apparent lack of population structure, *B. manusensis* is absent from Solwara 1, a site that occurs between Solwara 8 and South Su and that is within 2.5 km of South Su. Further, Solwara 1 shares many vent-dependent and vent-associated species with Solwara 8 and South Su (*Coffey Natural Systems, 2008*; *Erickson, Macko & Van Dover, 2009*; *Thaler et al., 2011*; *Thaler et al., 2014*; *Plouviez et al., 2013*).

The relatively homogeneous distribution of both *COI* haplotype and microsatellite markers for *Bathymodiolus manusensis* within Manus Basin is consistent with high gene flow between Solwara 8 and South Su. Similar levels of gene flow were observed in *Ifremeria nautilei* (*Thaler et al., 2011*) and *Chorocaris* sp. 2 (*Thaler et al., 2014*), although in both cases, the species were also found at Solwara 1. A significant, negative Fu's $F_S$ is consistent with a recent, rapid expansion, a pattern also observed in other species examined from these sites (*Thaler et al., 2011*; *Plouviez et al., 2013*; *Thaler et al., 2014*). *COI* haplotype diversity is higher at South Su (0.59 compared to Solwara 8's 0.45; Table 2), however, there is no consistent pattern of microsatellite richness between the two sites, nor are unique alleles consistently identified at one site over the other. A potential alternate explanation could be that both sites were recently colonized by the same cohort, and though currently isolated, have not had enough time for significant differentiation to accumulate via genetic drift.

In more than ten years of exploration and environmental observations, consisting of at least four research campaigns, neither *Bathymodiolus manusensis* nor any other mussel in the genus *Bathymodiolus* has been observed at Solwara 1 (W Saleu, pers. obs., 2014). Visual surveys of the seafloor suggest that adequate substrate (hard basalt surrounding low temperature venting fissures) exists within the Solwara 1 site for *B. manusensis* to settle (A Thaler, pers. obs., 2008), although the fluid chemistry that might influence mussel recruitment has not been characterized for Manus vents. In a previous study, we identified a similar pattern of presence/absence among populations of *Munidopsis lauensis* at Solwara 8 and South Su (*Thaler et al., 2014*). One population of *M. lauensis* was found at Solwara 8 and South Su, but absent at Solwara 1, while a second population was restricted to samples from Solwara 1 (*Thaler et al., 2014*). We hypothesized that sweepstakes effects related to the survival and settlement of recruits at vent sites in Manus Basin was responsible for the observed population structure of *M. lauensis* and that time series sampling would reveal a stochastic, dynamic distribution of these populations throughout the basin (*Thaler et al., 2014*).

When compared with putative *Bathymodiolus manusensis* samples from other sites in Manus Basin as well as Lau Basin and offshore New Zealand, there is a similarly high affinity between Solwara 8, South Su, and PACMANUS samples, suggesting a larger, well-mixed Manus population. *B. manusensis* sequences from outside of Manus Basin were basal to all Manus samples, suggesting that the population within Manus Basin is younger than

those from surrounding regions, and that there is a greater retention of propagules within Manus Basin, resulting in less gene flow and increased isolation within Manus Basin. Similar patterns of isolation within Manus Basin were observed in *Ifremeria nautilei* and *Chorocaris* sp. 2 (*Thaler et al., 2011*; *Thaler et al., 2014*).

That *Bathymodiolus manusensis* shares the same pattern of presence/absence with one population of *Munidopsis lauensis* suggests that the apparent exclusion of certain species or populations from Solwara 1 may be the result of a consistent, species- and population-dependent, dispersal barrier, rather than stochastic recruitment events. Other "leaky" dispersal barriers have been observed for hydrothermal vent populations across the equatorial East Pacific Rise (*Plouviez et al., 2009*; *Plouviez et al., 2010*; *Vrijenhoek, 2010*), but those sites were separated by thousands of kilometers, whereas the Manus Basin sites are 2.5–40 km apart. To determine if there is a barrier restricting some, but not all, species (or populations) from recruiting to Solwara 1, we need to sample additional species to identify consistent patterns across multiple taxa and sample the same species at additional time points to establish if observed patterns are temporally stable. The alternative hypothesis that species and populations are adapted to particular environmental conditions that are not always present at a site remains plausible (and not mutually exclusive), especially given well-documented evidence for such circumstances in *Alviniconcha* species in Lau Basin (*Beinart et al., 2012*) and the lack of fluid chemistry data from evident and putative mussel habitats at Manus Basin vents.

## Implications for management strategies

The limited distribution of *Bathymodiolus manusensis* and of a *Munidopsis lauensis* population within Manus Basin underscores the potential complexity of connectivity and habitat availability within vent ecosystems and the value of comprehensive environmental baselines prior to the initiation of an extractive regime (*Collins, Kennedy & Van Dover, 2012*; *Collins et al., 2013*; *Thaler et al., 2014*; *Boschen et al., 2016*). It is possible that multiple mining events in Manus Basin could affect source–sink dynamics of *B. manusensis* and other taxa, resulting in regime shifts in vent communities of Manus Basin as has been noted in other marine ecosystems (*Scheffer & Carpenter, 2003*). A similar phenomenon was observed in Moorea coral reef communities, where persistent disturbance, caused, in this case, by invasive crown-of-thorn starfish resulted in permanent changes in community structure as opportunistic recruits occupied newly exposed ecologic niches (*Berumen & Pratchett, 2006*).

The potential for regime shifts, where species not present at the disturbance site but occurring at neighboring sites establish a foothold following anthropogenic impacts, creates a challenge for environmental management and mitigation programs. At the very least, there is a need to understand if such a regime shift constitutes a significant adverse impact that should trigger a management response. Managers need to understand the extent of local variation in population and community structure to anticipate cumulative impacts and ecological consequences of regime changes following disturbance.

## ACKNOWLEDGEMENTS

We thank Dr. Samantha Smith and Renee Grogan of Nautilus Minerals, the captain and crew of the *M/V Nor Sky*, the Canyon Offshore ROV team, and Rebecca Jones and Pen-Yuan Hsing for assistance with field sampling. We thank Bernard Ball for laboratory assistance and Clifford Cunningham for advice and consultation. Specimens of *Bathymodiolus manusensis* from Manus Basin collected for this work are the property of Papua New Guinea, held in trust by Nautilus Minerals, and loaned to Duke University for baseline studies for the Solwara 1 Project. We thank two anonymous reviewers for their comments and critiques.

### Funding

This research was funded by a contract from Nautilus Minerals to CLVD and by Duke University. WS was supported by a 2009 Traineeship from the Endowment Fund of the International Seabed Authority. JC was funded from Science Foundation Ireland (SFI 12/IP/1308). The funders had no role in study design, data collection and analysis, decision to publish, or preparation of the manuscript.

### Grant Disclosures

The following grant information was disclosed by the authors:
Nautilus Minerals.
2009 Traineeship from the Endowment Fund of the International Seabed Authority.
Science Foundation Ireland: SFI 12/IP/1308.

### Competing Interests

William Saleu was an employee of Nautilus Minerals, a deep-sea mining company, though he was not at the time this study was conducted. Nautilus Minerals has provided research funding, logistical support, and vessels for studies conducted at these hydrothermal vent systems. Andrew Thaler is the CEO of Blackbeard Biologic, USA. William Saleu is the Principle Investigator of BETA Scientific, PNG.

### Author Contributions

- Andrew D. Thaler conceived and designed the experiments, performed the experiments, analyzed the data, wrote the paper, prepared figures and/or tables, reviewed drafts of the paper.
- William Saleu performed the experiments, analyzed the data, wrote the paper, prepared figures and/or tables, reviewed drafts of the paper.
- Jens Carlsson conceived and designed the experiments, analyzed the data, contributed reagents/materials/analysis tools, wrote the paper, reviewed drafts of the paper.
- Thomas F. Schultz conceived and designed the experiments, performed the experiments, analyzed the data, contributed reagents/materials/analysis tools, wrote the paper, reviewed drafts of the paper.

- Cindy L. Van Dover conceived and designed the experiments, contributed reagents/materials/analysis tools, wrote the paper, reviewed drafts of the paper.

## DNA Deposition

The following information was supplied regarding the deposition of DNA sequences:
GenBank (accession numbers KF498731–KF498847).

## Data Availability

The raw data has been supplied as Data S1–S2.

## Supplemental Information

Supplemental information for this article can be found online at http://dx.doi.org/10.7717/peerj.3655#supplemental-information.

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
