# Peer review of "Population structure of Bathymodiolus manusensis, a deep-sea hydrothermal vent-dependent mussel from Manus Basin, Papua New Guinea"

_PeerJ, doi:10.7717/peerj.3655_

## Round 0.1 · original submission · Major Revisions

I have heard back from two reviewers regarding your manuscript. Both had mostly minor comments on your work. However, reviewer 2 has mentioned one large issue with your dataset, suggesting comparisons with other reported datasets or at least a consideration of them. Please consider this and the other small and helpful comments carefully when making your revision. If you follow the suggestions of reviewer 2, the revisions will be quite substantial, and therefore my decision is 'major revision'.

Reviewer 1 ·

Basic reporting

No comments.

Experimental design

No comments.

Validity of the findings

No comments.

Additional comments

This is a well-written paper that I had pleasure to read. The problematic is presented very clearly and addressed using the proper methodology. The sample size are huge, even kind of an overkill if you ask me; state-of-art statistics are used adequately, well understood and well interpreted.

To be honest, I had hard time finding anything prone to real criticism and that is why I think it is reasonable to accept the paper for publication in its current form. I have two comments though, which I wish to bring to the attention of the authors. Both of these comments are actually related.

My first comment is related to the huge sampling effort and methodological overkill deployed to address the main question of the paper, that is, connectivity across the 2 sites studied. Let us not forget indeed that we are talking about deep-sea mussels, which planktonic phase has been shown (in closely related species) to disperse over thousands kilometers! In the regard, did it really make sense to deploy such effort to test genetic structure over only 50 kilometers? I note however that you implicitly already addressed my question by stating in the introduction that the absence of intermediate population might suggest a physical barrier to dispersal. Why not indeed... I also understand that this study is a part of a bigger puzzle and that as such, this question had to be addressed anyways.

My second comment is related to the interpretation of (the absence of) genetic structure in terms of gene flow. The data presented *are consistent* with gene flow indeed, and again it totally makes sense with regard to the spatial scale of the study and the life cycle of the organism considered. But one may want to remember that they do not *prove* gene flow. Just consider the hypothesis that the two sites were actually colonised recently from a single source and then underwent local expansion (which is consistent with the estimated Fu's Fs by the way). Under that hypothesis, the two populations might have been totally isolated since they colonised both sites and the null/low FSTs reflect a retention of ancestral polymorphism rather than current gene flow. In other words, under the hypothesis of a recent colonisation and considering the virtually infinite effective population sizes, even without gene flow, the two populations may not have had enough time to accumulate differences through genetic drift. One way to test this hypothesis would be to use a model that enables telling apart ancestral polymorphism from gene flow, such as the "Isolation with Migration" model presented by Hey (2010). It may be worth digging into that issue for further studies that will address connectivity along hydrothermal vents in the Manus Basin.

Hey (2010) Isolation with Migration Models for More Than Two Populations. Molecular Biology and Evolution 27 (4): 905-920.

Reviewer 2 ·

Basic reporting

The authors carried out population genetic analyses for the two populations collected in two deep-sea hydrothermal vent fields in Manus Basin. The results seem to contribute to establish a conservation plan along the mining and are worth sharing. However, I would like to know why the authors focus on only two sites, Solwara 8 and South Su, because B. manusesnsis had collected in the other hydrothermal vent field in the Manus Basin (PACMANUS field). Miyazaki et al. (2004) obtained partial COI sequences of five specimens of B. manusensis from PACMANUS field, using the same primer set to this study (Accession No.: AB101431 – AB101434). When I carried out BLAST search for the sequences in the supplemental .fas file provided by the authors, the sequences from the PACMANUS population showed high similarity (99%), as well as the sequences of the individuals from Lau Basin (Accession No: AB257539, AB257541, AB257543) and from New Zealand (Accession No.: AB255739 – AB255742). These two lineages (Lau Basin and New Zealand) are clearly different from the ones that the authors mentioned in L. 46 – 47 of their manuscript, as there are at least two species of Bathymodiolus (B. brevior and B. manusensis) in Lau Basin and New Zealand, according to Figure 3 in Miyazaki et al. (2010). These facts may indicate that B. manusesnsis seemed to distribute not only in Manus Basin, but also in wider area in the southwestern Pacific. I would suggest the authors to reconsider their hypothesis and background of this study with reviewing the reported datasets (at least listed above) and papers (listed below).
Miyazaki J, Shintaku M, Kyuno A, Fujiwara Y, Hashimoto J, Iwasaki H (2004) Phylogenetic relationships of deep-sea mussels of the genus Bathymodiolus (Bivalvia: Mytilidae). Marine Biology, 144: 527 – 535.
Miyazaki J, Martins LO, Fujita Y, Matsumoto H, Fujiwara Y (2010) Evolutionary process of deep-sea Bathymodiolus mussels. PLoS One, e10363.

Point-to-point comments;
L. 85: A parenthesis is not closed.
L. 94: Folmer’s original paper described the names of primers as “LCO1490” and “HCO2198”, which were not italicized (because these are the names for primer, not for gene).
L. 97: The word “Reactions” may be replaced by “Products”.
L. 156: The word “Bathymodiolus” can be abbreviated as “B.” here, as well as in L. 162,174, 191, 201, 213.
L. 159: The word “CO1” must be replaced by italicized “COI”.
L. 165: “P” should be italicized.
L. 194: “Fs” should be italicized.
L. 214: The word “Munidopsis” can be abbreviated as “M.” here.

Experimental design

Point-to-point comments;
L. 95: Company which provided Taq polymrase is not described.
L. 103: “Big Dye” must be “BigDye” (space between “Big” and “Dye” is not required).

Validity of the findings

Point-to-point comments;
L. 203 – 205: It is difficult for me to follow this suggestion. How did the authors know that B. manusensis could settle in Solwara 1, without any observation of Bathymodiolus mussels in this site (L. 201 – 202)?

Additional comments

No comments

---

## Round 0.2 · accepted · Accept

The authors have revised the submission very well, and addressed all concerns of both reviewers adequately. I look forward to seeing the published version of this work!